



# Using Multi-Head Attention Deep Neural Network for Bias Correction and Downscaling for Daily Rainfall Pattern of a Subtropical Island

Yi-Chi Wang[1], Chia-Hao Chiang[1], Chiung-Jui Su[1], Ko-Chih Wang[2*], Wan-Ling Tseng[1], Cheng-Ta Chen[3], Hsin-Chien Liang[1]

[1]Research Center for Environmental Changes, Academia Sinica, Taipei, Taiwan
[2]Department of Computer Science, National Taiwan Normal University, Taipei, Taiwan
[3]Department of Earth Sciences, National Taiwan Normal University, Taipei, Taiwan

*Correspondence to*: Ko-Chih Wang (kcwang@ntnu.edu.tw)

Abstract. This study investigates the capability of a deep learning approach, employing a multi-head attention mechanism within a deep neural network (DNN) framework, aimed at refining the bias correction and downscaling process for the fifth generation European Centre for Medium-Range Weather Forecasts reanalysis rainfall datasets to provide local-scale daily rainfall data across Taiwan, a mountainous subtropical island. Leveraging gridded 5-km daily rainfall observations across Taiwan, the proposed DNN model, the Encoder-Decoder with multi-head Attention for auxiliary channels (EDA) model, can adeptly correct biases and downscale rainfall statistics from coarse-resolution reanalysis data by incorporating auxiliary inputs, such as surface wind information, and invariant data, such as high-resolution topography data. Our evaluation, centred on the distinct seasonal rainfall characteristics of Taiwan, uses mean rainfall patterns, rainfall statistics, extreme climate indices, and their interannual variation for the rainy seasons. The findings show the EDA model's ability to correct for overestimated low-intensity rainfall and inaccurately positioned orographic rainfall in reanalysis datasets, achieving better accuracy than conventional quantile-mapping methods. Further analysis reveals the critical role of auxiliary information of surface winds used by the EDA to enhance the downscaling accuracy across various performance metrics. This study underscores the significant potential of DNN architectures for statistical bias correction and downscaling in regions with complex terrains, by effectively integrating auxiliary data to capture the interplay between synoptic and local circulations influenced by topography.

**Keywords:** climate downscaling; bias correction; deep neural network; multi-head attention layer; orographical rainfall; Taiwan

## 1 Introduction

The rising frequency and intensity of extreme events, particularly heavy rainfall, underscore the critical need for localized and applicable climate predictions to mitigate their impacts on society. Global Climate Models (GCMs) and Earth System



Models (ESMs) are instrumental in forecasting changes of extreme events in future climate scenarios, based on varied
projections of radiative forcing and human influence. However, these models generally operate with a spatial resolution around
100 kilometres, which is insufficient for detailed assessments of climate risks and the development of effective adaptation and
mitigation strategies for local communities. To address the gap, the process of downscaling is often employed for refining
coarse-scale information from GCMs and ESMs into the fine-scale statistics of critical surface variables, such as rainfall and
temperature, thereby making them applicable to localized contexts.
The field of climate downscaling is currently dominated by two primary methodologies: dynamical and statistical
downscaling (Maraun et al., 2010). Dynamical downscaling leverages high-resolution regional climate models to simulate
local-scale climate variability, proving particularly beneficial in capturing extreme rainfall events when higher resolution
models are employed (Westra et al., 2014). Despite its advantages, the dynamical downscaling method demands significant
computational resources and struggles to encapsulate uncertainties across extensive ensemble simulations. Statistical
downscaling, conversely, constructs empirical relationships between coarse-resolution variables from GCMs and local-scale
surface variables, helping the application of climate projections at a more granular level (Maraun and Widmann, 2018). Among
them, the Model Output Statistics (MOS) method is particularly valued for its straightforward approach, requiring no prior
knowledge for the selection of predictors or regions. It utilizes GCM outputs directly to train statistical models, emulating the
relationship between model outputs and observational data. This method primarily focuses on adjusting the rainfall distribution
to align more closely with observations, thereby correcting systematic errors found in GCMs. However, limitations and
disparities exist across various MOS techniques, each with unique strengths and weaknesses (Soares et al., 2019; Maraun and
Widmann, 2018; Vogel et al., 2023). Thus, the selection of downscaling and bias correction methods needs a comprehensive
understanding of the specific climate phenomena and the capabilities of the parent models (Maraun et al., 2017).
Recent advancements in deep learning (LeCun et al., 2015), particularly Convolutional Neural Networks (CNNs; Lecun
et al., 1998), have garnered attention in climate science due to their success in finding patterns within data, paralleling tasks in
climate research field (Reichstein et al., 2019). Inspired by super-resolution techniques in computer vision, which enhance
image detail from low-resolution inputs (Dong et al., 2014), applications in climate science have demonstrated the potential of
deep learning models to refine spatial resolution on pure-resolution approaches, which utilizing coarsen version of high-
resolution ground truth data to train their model, like the DeepSD model by Vandal et al., (2017), and other variants for regions
like India (Kumar et al., 2021) and southeastern United States (Wang et al., 2021), as well as the continental United States
(Sha et al., 2020) . However, these studies primarily addressed the upscaling aspect, leaving room for improvement in bias
correction. Addressing the comprehensive challenge of both upsampling and bias correction, the literature reveals a diverse
array of deep neural network (DNN) applications. Notably, the integration of skip connections within the encoder-decoder
architecture, as seen in the YNet model developed by Liu et al. (2020), is a significant advancement. This model shows
enhanced efficiency and flexibility over DeepSD by incorporating orographic data, thus addressing daily rainfall statistics
downscaling from GCMs with 100 km resolution across the Continental United States. The versatility of DNN architectures



enables the exploration of using various climate variables and their interactions as the input for downscaling tasks, including the univariate rainfall variables (Liu et al., 2020; Vandal et al., 2017; Wang et al., 2021; Rocha Rodrigues et al., 2018; Saha and Ravela, 2022), with surface variables (Oyama et al., 2023; Sun and Tang, 2020), with covariance between rainfall and free-troposheric variables (Wang et al., 2023), and multiple atmospheric fields (Harris et al., 2022; Price and Rasp, 2022; Baño-Medina et al., 2021, 2020; Adewoyin et al., 2021; Sun and Lan, 2021). This broad spectrum of research underscores the profound potential and flexibility of DNNs in tackling the intricate problems of climate downscaling, offering paths forward in both resolution enhancement and bias correction.

This study would like to take the advantage of DNNs to enhance the downscaling and bias correction process, particularly addressing the challenge of orographic rainfall bias. This bias, a common issue in regions with complex terrain, results from the misplacement of rainfall in GCMs due to insufficient grid resolution to accurately model local circulations affected by orographic lifting and the biases arising from the physical parameterization of rainfall processes (Maraun and Widmann, 2015; Cannon et al., 2015). Taiwan's topography is marked by a series of major mountains reaching elevations of up to 4,000 meters, stretching in a north-south direction across a compact longitudinal span of 200 kilometres. This unique geographical setting makes the island an ideal site for our research (Fig. 1). The intricate landscape results in pronounced geographical variations, shaped by the interplay between East Asian monsoonal flows and the island's topography. Through extensive observational studies, the rainfall seasons in Taiwan have been categorized, taking into account the dominant rainfall systems within the monsoon. Especially in summer, the Meiyu frontal system and typhoons greatly shape the seasonal rainfall and the patterns of extreme rainfall in Taiwan. Such detailed understanding provides a comprehensive framework for exploring the effectiveness of downscaling methods under different rainfall regimes (Chen and Chen, 2003; Henny et al., 2021).

Drawing inspiration from the YNet model and its incorporation of attention mechanisms to predict daily rainfall patterns over Taiwan, Chiang et al. (2024) demonstrated the advantages of including bias correction and downscaling components within their DNN model, noting improved performance in terms of prediction skills, RMSE, and correlation. However, their approach to dataset partitioning through random choice was identified as suboptimal for climate downscaling applications, which are typically oriented towards future projections. Additionally, their model showed a tendency to prioritize the prediction of weak rainfall events over more extreme rainfall events, likely a consequence of employing mean square error (MSE) as the loss function. Building on the insights of their work, our study introduces an advanced DNN model, the Encoder-Decoder with multi-head Attention layers for auxiliary channels (EDA), designed to enhance feature extraction capabilities through the inclusion of multiple auxiliary channels. We also propose a revision to the loss function, adopting weighted MSE to better capture the nuances of extreme rainfall events. Furthermore, we have refined the training and validation procedures by opting for partitioning based on consecutive years, ensuring a more suitable approach for the specific needs of climate downscaling applications. This methodological enhancement aims to more accurately model and predict the intricate patterns of rainfall in Taiwan, addressing both the challenges of orographic rainfall bias and the broader demands of climate downscaling.



To address our research goal, we have designed a training and evaluation framework that aligns with the principles of
univariate downscaling of rainfall within the MOS framework, incorporating surface winds and topography as auxiliary
datasets. We employ the fifth-generation European Centre for Medium-Range Weather Forecasts (ECMWF) reanalysis
(ERA5; Hersbach et al., 2020) dataset, with a spatial resolution of 0.25°, as the input data, and use local-scale gridded
observations in Taiwan with a 5km resolution as the ground truth. This approach primarily targets model biases—specifically
those arising from the coarser resolution of model grids and the rainfall parameterization—rather than biases associated with
the large-scale environmental representations in GCMs. Our methodology is tailored for future climate downscaling rather
than nowcasting, with a specific focus on daily temporal resolution to enhance the relevance of this approach for climate-
related applications.
The upcoming sections of this paper are structured as follows: Session 2 will outline the methodology employed in this
study, detailing the data sources for training and evaluation, along with the architecture of the proposed DNN model. Session
3 will present a comparative analysis of the BCSD and EDA models, focusing on their ability to simulate mean rainfall patterns,
extreme rainfall indices, rainfall statistics of selected observational stations, and interannual variation of extreme indices across
five distinct seasonal rainfall regimes. Session 4 will give a summary of our results and delve into the broader implications for
future research and practical applications in the field of climate downscaling.
**2 Methodology**
**2.1 Data**
In this research, our downscaling model utilizes daily rainfall data from two key sources: the fifth-generation European
Centre for Medium-Range Weather Forecasts (ECMWF) reanalysis dataset (ERA5; Hersbach et al., 2020), which offers
coarse-resolution input with 25 km grids, and the high-resolution observed gridded daily rainfall data for Taiwan. The latter is
provided by the Taiwan Climate Change Projection Information and Adaptation Knowledge Platform (TCCIP; Lin et al.,
2022), featuring 5 km grids. This setup allows us to test the model's efficacy by using ERA5's broader-scale data as input to
predict more localized rainfall patterns, with the TCCIP data serving as a high-resolution ground truth for validation.
The ERA5 dataset, developed by the ECMWF, merges cutting-edge global weather modelling with an extensive array of
observational data through sophisticated data assimilation techniques. Although its underlying modelling resolution stands at
9 km, ERA5 provides atmospheric variables, including rainfall, at a coarser 25 km horizontal resolution. This dataset is noted
for its overall reliability in rainfall monitoring against observational networks. However, its broader resolution and the inherent
biases from its parameterization approach can introduce discrepancies, particularly in areas where the interplay between
synoptic weather patterns and topography is pronounced (Rivoire et al., 2021). To capture these complex interactions, our
model, EDA, incorporates ERA5-derived daily aggregated rainfall and 10-meter height wind data as inputs. This daily





aggregation is meticulously compiled from ERA5's hourly rainfall data, ensuring a detailed representation of daily rainfall patterns for our analysis.

The ground truth TCCIP dataset is constructed from observations collected by a comprehensive network of 2,203 stations across Taiwan. These stations, operated by various Taiwanese agencies such as the Central Weather Administration, Civil Aeronautics Administration, and others, contribute to a rich dataset that has facilitated extensive climate studies, including extreme rainfall trend analysis (Tung et al., 2022; Henny et al., 2023, 2021). Objective analysis, employing Gaussian latent variables, is applied to transform station-based measurements into the gridded format (Weng and Yang, 2018). Our study uses data spanning from 1960 to 2020. The resolution of 5 km in TCCIP dataset is well-suited for county-scale climate impact assessments in Taiwan.

Our study zeroes in on the geographical area of Taiwan (22°N-25°N, 120°E-122°E) as delineated in the ERA5 dataset, with a specific focus on the island's land regions for the purpose of training and validating our models against the TCCIP dataset. To enhance our analysis, we integrated a topographical dataset for the Taiwan region, provided by the GIS centre at Academia Sinica, Taiwan. This dataset uses the Terra Advanced Spaceborne Thermal Emission and Reflection Radiometer (ASTER) Global Digital Elevation Model (GDEM), a collaborative product from Japan's Ministry of Economy, Trade, and Industry and NASA (NASA/METI/Japan Space systems and US/Japan ASTER Science Team, 2019). Originally detailed at a 20-meter resolution, this topography data was regridded to a 5-km resolution to better match the TCCIP rainfall data's resolution, facilitating a more aligned analysis.

## 2.2 Training and Validating Procedure

Statistical downscaling fundamentally aims to establish relationships between the expected value of local-scale predictands Y, based on large-scale predictors X, as outlined by X (Maraun et al., 2010). This relationship can be expressed as follows.

$$E(Y|X) = f(X, \theta)$$

where $\theta$ represents adjustable parameters within the downscaling framework. In our study, Y denotes the TCCIP rainfall data of 5 km, and X refers to the ERA5 rainfall data of 25 km.

For the training, validation, and testing phases, we segmented the data into three distinct periods. The training dataset spans from 1960 to 2014, the validation dataset covers the years 2015 to 2017, and the test dataset encompasses the period from 2018 to 2020. With a total of 22,281 daily precipitation records, the data is divided such that 80% is allocated for training, 10% for validation, and the remaining 10% for testing. This separation into distinct sets for testing and validation enables us to more accurately assess the model's predictive uncertainties across varying data regimes. Our choice of temporal division is designed to mimic typical practices in climate science, aiming for forecasts of future climate changes in a sequential manner rather than employing the random splitting commonly used in data science fields.



The training protocol for our model includes a series of preprocessing steps designed to optimize the input data for effective learning. These steps encompass a log1p transformation to adjust for the skewness in the distribution of the data values, particularly beneficial for precipitation data. Moreover, we normalize various data variables to ensure consistency across the dataset: precipitation, temperature, and humidity data are normalized to a [0,1] range, whereas wind vector data at 10 meters height is normalized to a [-1, 1] range. This normalization strategy facilitates the model's learning process by enhancing convergence rates, promoting generalization capabilities, optimizing performance, and reducing the model's sensitivity to initial parameter settings. These improvements collectively contribute to an increase in the model's computational efficiency and predictive accuracy.

### 2.3 Model Structure: Encoder-Decoder with multi-head Attention layers for auxiliary channels (EDA)

The proposed model here, termed the Encoder-Decoder with multi-head Attention (EDA), evolves from the framework established by Chiang et al. (2024), comprising two main components: an encoder and an encoder. The innovation in our model primarily lies within the encoder, where we have replaced traditional convolutional layers with multi-head attention layers and fully connected layers derived from the Transformer architecture (Vaswani et al., 2017). Unlike conventional neural networks that rely on recurrent or convolutional layers, the Transformer architecture is built entirely around attention mechanisms, facilitating direct modelling of dependencies regardless of their distance in the input data. This capability is pivotal for our model, allowing it to simultaneously process the entire dataset and enabling each grid point to evaluate its relationship with all others. This approach not only captures the intricate interdependencies characteristic of climate variables but also introduces flexibility in handling input data of varying sizes. By projecting inputs into a feature space where the attention mechanism operates, the model accommodates a broader range of auxiliary data from the climate system, enhancing its adaptability and applicability.

Figure 3 depicts the architecture of the EDA, showcasing the neural network's hidden layers. The encoder plays a crucial role in extracting representative features and patterns from climate variables, as well as understanding the spatial relationships among grid points. This is accomplished by initially transforming the input data—comprising flattened, multi-variable climate information—into high-dimensional vector representations. Subsequently, the encoder utilizes a multi-head attention mechanism to uncover latent patterns within these vectors, where the diversity of patterns detected is directly proportional to the number of attention heads employed. Through this mechanism, the model effectively identifies and emphasizes areas of significant correlation or importance across the grid, enabling each attention head to capture unique facets of the data's structure at lower resolution.

In the design of the decoder component of our model, we have maintained a CNN structure. Decoder part is designed flexibly that one could implement the desired sub-model for combining the intermediate outputs from the encoder with the topography data and performing a one-step upscaling. As for the downscaling process, the intermediate outputs from the encoder are transitioned to the decoder, which are initially reshaped into two-dimensional gridded data before being processed





by the decoder. This step ensures the model to rearrange and to reform the spatial relationships between data points and
prepares for concatenating with topography data also for achieving higher resolution. In our case, we have adopted Image
Super-Resolution Using Deep Convolutional Networks (Dong et al., 2014) and Enhanced Super-Resolution CNN (Shi et al.,
2016) to capture the non-linear mapping to high-resolution rainfall.
For our downscaling task, it is achieved through a one-step upscaling layer using pixel shuffling, an interpolation
technique from Enhanced Super-Resolution CNN (Shi et al., 2016), which, when combined with geographical data, enables
the model to learn the complex interactions between precipitation and elevation, such as orographic rainfall effects. Together,
these elements enable the decoder to meticulously process and enhance the data, ensuring the generation of detailed and
accurate high-resolution climate predictions. This approach significantly contributes to the local interactions between
topography and feature maps, aiding in the precise downscaling of climate data.
Implementation is carried out within the TensorFlow framework, leveraging its robust capabilities for efficient model
training and optimization. The training batch size is set to 64, and the training duration is capped at a maximum of 1,000
epochs, incorporating an early stopping mechanism activated if there is no improvement in the loss function for 60 consecutive
epochs. Our model employs a weighted mean square error (WMSE) as the loss function as follows:
$$WMSE = \frac{1}{HW} \sum_{i=1}^{H} \sum_{j=1}^{W} [\gamma \widehat{Y_{ij}} (\widehat{Y_{ij}} - Y_{ij})^2 + (1 - \gamma)(\widehat{Y_{ij}} - Y_{ij})^2], \gamma \in [0,1],$$

where H and W are the height and width, $Y_{ij}$ is the prediction and $\widehat{Y_{ij}}$ is the corresponding ground truth. This approach allows
for the imposition of greater penalties on errors in regions characterized by high rainfall, addressing the critical need for
accuracy in predicting extreme weather events. The training regime is executed in a supervised manner, with an initial focus
on training the encoder using low-resolution observational data. Subsequent to this phase, the encoder is frozen, and the
encoder is trained on high-resolution data, a strategy designed to fine-tune the model's ability to perform accurate downscaling
and bias correction.
Optimization is achieved through the use of the Adam Optimizer, set with a learning rate of $10^{-4}$, to adjust model
parameters effectively during the training process. Training is performed on an NVIDIA® Tesla V100 GPU, a choice that
significantly enhances computational efficiency, allowing the entire training process to be completed in just over 10 hours.
This setup ensures that the model is both accurately and efficiently trained to meet the demands of precise climate data
downscaling.

**2.4 Baseline Downscaling Methods: Bias Correction Spatial Disaggregation method (BCSD)**

For benchmarking within the univariate downscaling framework in our study, we have adopted the Bias Correction Spatial
Disaggregation (BCSD) method as our comparative baseline. Developed by Wood et al. (2002), BCSD merges spatial and
temporal disaggregation for downscaling with a quantile mapping (QM)-based technique for bias correction, designed to align
the modelled data distribution with the observed distribution over corresponding periods effectively. This method is notable





for its capability to maintain the mean percentile of data distribution efficiently, its computational effectiveness, and its
independence from requiring prior specific information, making it widely used in regional studies (Bürger et al., 2013; Cannon
et al., 2015; Maraun et al., 2010), including that focused on Taiwan by TCCIP (Lin et al., 2023). However, BCSD, as a
representative QM downscaling method, shares common challenges associated with QM-type methods, including potential
shifts on the tails of the distribution, the incapability to correct misrepresenting location bias in coarse-resolution datasets
(Maraun and Widmann, 2018; Maraun et al., 2017; Maraun and Widmann, 2015), as well as the challenge in preserving long-
term climate trends within the data (Cannon et al., 2015).
In our current implementation of BCSD, we have omitted the original design of the temporal disaggregation step for
converting monthly-resolved data into a daily time scale and adopt a step with daily rainfall data in line with the methodologies
of recent studies (Thrasher et al., 2012; Vandal et al., 2019). For other details, we adhere closely to the methodological
framework for statistical downscaling in Taiwan with CMIP6 models, as outlined by Lin et al. (2023), including the 3 steps
below:
1. First, the ERA5 rainfall data is bilinearly interpolated onto the TCCIP data grid, transitioning from 25km to 5km

grids to align with the TCCIP dataset's fine-scale resolution.

2. Subsequently, employing a 31-day time window centred around the target day for each grid point, we construct the

cumulative distribution function (CDF) in a manner that effectively captures the climatological distribution, using

61 years of TCCIP gridded rainfall data. A bin width of 15%, determined through empirical testing, is applied in

constructing the CDF for each grid point.

3. The final step is to adjust the interpolated coarse-resolution rainfall data to the observational rainfall's corresponding

CDF quantiles using the QM method.

**2.5 Evaluation Metrics**
Here, we have listed model metrics that we used in understanding the performance of downscaling methods. We first
examine the season mean over the 5 raining seasons in Taiwan and evaluate the performance of spatial pattern based on
Pearson's correlation (CORR) and root mean square error (RMSE) of spatial pattern over Taiwan.
To quantify the performance of rainfall extremes of downscaling methods, we have also used the extreme indices
developed by the joint Expert Team on Climate Change Detection and Indices (ETCCDI) of the WMO Commission for
Climatology and World Climate Research Programme Climate Variability and Predictability project (Karl et al., 1999; Frich
et al., 2002; Zhang et al., 2011), which are widely used in many studies about extreme events, including in several IPCC reports
(Sillmann et al., 2013). For the following definition of extreme indices, we have let $RR_{ij}$ as the daily rainfall amount on day i
in period j. Then the extreme indices defined as follows:
● RX1day (Monthly maximum 1-day precipitation): $RX1day_j = \max(RR_{ij})$ for a period j.



- CDD (Maximum length of consecutive days with RR<1mm): The largest number of consecutive days when $RR_{ij} < 1$ mm of each day i in period j.
- SDII (Simple precipitation intensity index): Given wet days defined as days with RR > 1 mm in period j and W as the number of wet days in period j, the $SDII_j$ is defined as $SDIIj = \sum_{w=1}^{w=W} RR_{wj}/W$.
- RX10mm (annual count of days when $RR_{ij} \geq 10$ mm): The number of rainy days when daily $RR_{ij} > 10$ mm in period j.

A complete set of climate indices used in the observations are listed in Appendix.

## 3 Results

### 3.1 Seasonal Rainfall Mean for 5 Seasons in Taiwan

Figure 1a illustrates the topographical contours of Taiwan, utilizing the ASTER GDEM dataset to delineate the elevation. We especially emphasize two predominant mountain ranges, Xue Mountain (XM) and Central Mountain (CM), on the figure. These ranges, oriented from the south to the north, exhibit elevations exceeding 2000 meters, can strongly interact with synoptic systems to have critical impact on rainfall patterns across the island.

Figure 1b delineates the annual rainfall cycle of Taiwan using daily data from the TCCIP, highlighting the distinct seasonal variations in rainfall across climatology, test, and validation periods. Taiwan's rainfall distribution exhibits five clear seasons: spring, the first and second rainy seasons, autumn, and winter, each closely associated with the East Asian (EA) monsoon system. Season-specific rainfall patterns, as identified in prior climatological studies and summarized in Table 1, mark each of these periods (Chou et al. 2009). Climatologically, rainfall intensifies in March with the onset of spring, escalating to an average of 15 mm/day across the island during the first and second rainy seasons of summer (illustrated by the black line in Fig.1b). However, notable fluctuations are observed during these rainy seasons, with daily extreme rainfall exceeding 30 mm/day during the passage of the Meiyu front in the first rainy season, and reaching up to 50 mm/day during typhoon or low-pressure system activities in the second rainy season (depicted by red and purple lines in Fig.1b). This pronounced variability underscores the dynamic nature of Taiwan's rainfall patterns across its distinct seasons.

Figure 1c presents the spatial distribution of mean rainfall across Taiwan's five distinct rainy seasons in climatology. In spring, as the subtropical high over the northwest Pacific shifts north-westward, cold frontal systems introduce rainfall to northwestern Taiwan, particularly affecting the southern slopes of Xue Mountain (Fig.1b). Summer in Taiwan is characterized by two distinct peak rainfall periods, known as the first and second rainy seasons, driven by monsoonal south-westerly flows that carry moisture from the tropics. The first rainy season sees significant rainfall, with daily averages up to 30 mm, especially on the southwestern slopes of the Central Mountain (Fig. 1c). Additionally, a prominent rainfall hotspot forms in the central western part of Taiwan, a continuation of the spring rainfall pattern, largely due to Meiyu frontal systems. These systems, extended east-west bands of rain, are noted for their mesoscale convective activity. The second rainy season is defined by





typhoon-driven rainfall, enhancing the moisture brought by the south-westerly flow. Typhoons, emerging from the tropical
Pacific and advancing from the east, deliver substantial rainfall to the eastern slopes of the central mountain (Fig. 1d). As
autumn arrives, the monsoon circulation alters, with prevailing winds becoming northeasterly and increasing rainfall on eastern
Taiwan's windward slopes (Fig. 1e). During winter, the focus of rainfall shifts to northeastern Taiwan, marking a seasonal
transition in precipitation patterns (Fig. 1f).
Figure 3 compares the seasonal mean rainfall across the five seasons between ERA5 reanalysis, BCSD, and our EDA
model. The ERA5 reanalysis demonstrates a notable displacement in the spring, inaccurately positioning the maximum rainfall
over eastern Taiwan (Fig.3a). Both BCSD and EDA model successfully correct this bias, realigning the maximum rainfall to
northwestern Taiwan where local orography enhances upslope rainfall, with a bias residual of less than 2 mm/day (Fig.3b, 3c).
The 1$^{st}$ wet season poses challenges for ERA5, which overestimates rainfall on the eastern side and near the western coast of
Taiwan (Fig.3a). In contrast, observed TCCIP rainfall predominantly occurs over the southwestern foothills of the Central
Mountain (Fig.1). Both BCSD and EDA model adjust this bias, redirecting rainfall to the southwestern part of Taiwan,
particularly over the southwestern foothills of the Central Mountain. The EDA model, however, shows a superior performance,
capturing the rainfall magnitude of 15 mm/day along the mountain and peaking at 20 mm/day at the southern tip of Taiwan.
In the second wet season in summer, while ERA5 accurately locates the southwestern rainfall maximum, it overestimates
rainfall in northern Taiwan (Fig. 3a). Both BCSD and EDA model implement crucial adjustments, effectively delineating the
contrast between the drier northern Taiwan and the wetter southwestern Taiwan (Fig.3b, 3c). During the autumn and winter
seasons, ERA5 predicts excessive rainfall on the eastern side of the Central Mountain (Fig.3a), in contrast to observed rainfall
hotspots that are predominantly located on the windward side of the Central Mountain. (Fig.1) BCSD and EDA demonstrate
comparable skill in amending this bias in these seasons (Fig.3b, 3c), attributed to ERA5's enhanced ability to depict the rainfall
pattern during winter, which aligns more closely with the moisture inflow associated with the northeasterly monsoonal flow,
typically resulting in a more uniform rainfall distribution.

## 3.2 ETCCDI Extreme Indices

We analysed climate extreme indices as recommended by the Expert Team on Climate Change Detection and Indices
(ETCCDI) using rainfall data from TCCIP, BCSD, and our EDA model. This analysis includes the maximum 1-day rainfall
(RX1day; Fig.4), the number of days with intense rainfall exceeding 10 mm (RX10mm; Fig.5), and the longest stretch of
consecutive dry days (CDD; Fig.6). Additional results for other ETCCDI indices are presented in the Appendix. In the
meantime, the CORR and RMSE of each model and TCCIP observations are summarized in Table 4 for the ETCCDI indices.
Figure 4 illustrates the climatological distribution of maximum 1-day rainfall (RX1day) across five rainy seasons within
the test period. During spring, the maximum of RX1day is predominantly observed over the northwest and on the southern
edge of Xue Mountain (approximately 24˚N-24.5˚N), with RX1day values reaching up to 60 mm/day. In the summer's wet
seasons, RX1day values exceed 100 mm/day, with peaks up to 300 mm/day observed on the southwest slope of the Central





Mountain. The magnitude of RX1day over the Xue Mountain diminishes gradually, with rainfall shifting towards the eastern
slope of the Central Mountain, where it is influenced by typhoon-related rainfall from the tropical Pacific (Fig.4a). Both BCSD
and EDA models exhibit comparable performances in spring, fall, and winter—seasons characterized by less extreme rainfall
events. During the two wet seasons in summer, notably, both models tend to underestimate the extreme rainfall over the
southwest slope of the Central Mountain, with discrepancies of up to 50 mm/day in summer seasons (Figs. 4b, 4c). Nonetheless,
the EDA model displays a more random-like distribution of RX1day across southwest Taiwan, compared with the dry bias
over the southwestern Taiwan indicating a wider dry bias. Furthermore, EDA demonstrates a better capability to capture the
rainfall over the eastern side of Taiwan in 1st wet season and northern Taiwan in Fall, compared to BCSD, suggesting a more
accurate representation of rainfall extremes during the typhoon seasons (Figs. 4b, 4c).
Figure 5 displays the spatial distribution of days experiencing rainfall exceeding 10 mm (RX10mm) as recorded by TCCIP
and contrasts the predictive discrepancies between BCSD and EDA during the test period. RX10mm is a critical measure for
identifying days characterized by significant rainfall. Consistent with the overall mean rainfall distribution, the bulk of rainy
days is concentrated during the first and second summer seasons, notably on the western slopes of the Central Mountain.
Rainfall in the northeastern and eastern parts of Taiwan begins in fall and continues through winter (Fig.5a). Throughout the
five seasons, BCSD tends to slightly underestimate the frequency of rainy days, a tendency mirrored by the EDA model.
Notably, during the first wet season, BCSD shows a marked underestimation, missing rainy days by up to 10 days across the
central mountains, particularly near the southern tip of Xue Mountain. This discrepancy arises as the ERA5 reanalysis
inaccurately captures rainfall locations during the first wet season, challenging BCSD's ability to identify significant rainfall
events despite its effectiveness in adjusting mean rainfall levels (Fig.5b). A similar pattern of underestimation by BCSD is
observed for RX10mm hotspots in the northeastern part of Taiwan during fall and winter. However, the EDA model manages
to mitigate BCSD's dry bias to a considerable extent, though it still portrays a drier Yilan region compared to observations
(Fig. 1a).
Figure 6 delineates the spatial distribution of consecutive dry days (CDD) throughout the five rainy seasons. According
to observations, CDD typically averages about 10 days during the rainy seasons of spring and summer, with a peak in fall
across western Taiwan. This trend continues into winter when CDD can extend up to 20 days in southern Taiwan (Fig.6a). The
bias exhibited by BCSD varies across different seasons (Fig.6b). In spring, BCSD appears to overpredict rainfall in southern
Taiwan, which results in an underprediction of CDD. During the summer's wet seasons, BCSD consistently overestimates
CDD throughout Taiwan. In fall, BCSD's predictions overestimate CDD in northeastern Taiwan and similarly overestimate
CDD in western Taiwan. Winter, generally a dry season for western Taiwan, sees BCSD overestimating CDD in west-central
Taiwan while underestimating it in the southwest. By contrast, the EDA model demonstrates a markedly lower bias in spring,
summer, and winter, more closely aligning with the observed CDD patterns. However, it tends to underestimate CDD during
fall, indicating a nuanced yet imperfect prediction capability for dry periods throughout the seasons. Comparisons using CORR
(correlation coefficient) and RMSE (root mean square error) metrics further underscore the EDA model's superior performance





across all seasons, with the notable exception of the second wet season during summer (Table 4). The challenge in accurately modelling this season may stem from the substantial contribution of typhoon-related rainfall, which, due to its somewhat stochastic nature compared with other seasons, complicates the precise prediction of rainfall distribution.

### 3.3 Rainfall Statistics of CWA Stations

The Central Weather Administration (CWA) of Taiwan operates an extensive network of observation stations across the island, situated in densely populated areas and critical topographical points. Our analysis focused on rainfall data from the three selected CWA stations, Tainan, Taichung, and Alishan, showcasing the climatological rainfall statistics during the two summer rainy seasons (Fig.7). Tainan and Taichung, located in southern and central Taiwan respectively, represent two urban cities on the plains, whereas Alishan is positioned on mountain slopes at an elevation of 1500 meters, providing insights into the impact of elevation on rainfall patterns. During the first wet season of summer, our analysis highlights a consistent issue with the ERA5 reanalysis: the overprediction of low-intensity rainfall events (less than 5 mm/day) across all stations (Fig.7a, 7b). This pattern illustrates the constraints of coarse-resolution models like ERA5, which tend to miss capturing extreme rainfall events and favour the forecasting of more frequent, yet milder, rainfall. In contrast, the EDA model shows considerable improvement in mitigating this bias for overly frequent, weak rainfall (under 5 mm/day). Additionally, the EDA model more accurately adjusts for heavy rainfall events (over 50 mm/day), especially at the Alishan station, a location significantly influenced by orographic rainfall. Here, the EDA model's performance is notably superior compared to the other stations.

For the second wet season, rainfall distribution among the urban stations, Tainan and Taichung, becomes more consistent, yet the EDA model maintains its precision in correcting both low (< 5 mm/day) and high (> 50 mm/day) rainfall categories (Fig.7a, 7b). Like in the first wet season, the EDA model's corrections are particularly effective at the Alishan station, successfully addressing the wet bias noted in the ERA5 reanalysis. On the other hand, the BCSD method tends to excel in adjusting rainfall within the mid-range spectrum, around 10 mm/day, accurately reflecting the average rainfall percentile for all three stations. However, it tends to overestimate the frequency of weak rainfall events even with correction from ERA5 statistics.

In summary, while the BCSD method adeptly adjusts mid-range rainfall amounts, the EDA model stands out for its ability to accurately correct rainfall across the spectrum, significantly improving the representation of both minimal and intense rainfall events. This distinction underlines the EDA model's capability to address biases in rainfall statistics, showcasing its effectiveness in capturing the complexities of rainfall patterns across different terrains and seasons.



**3.4 Interannual Variations of Mean Rainfall and Rainfall Extremes**

We examined the interannual variation in mean rainfall and extreme indices for the summer seasons, utilizing data from TCCIP, ERA5, BCSD, and the EDA model, as depicted in Figure 8. Our analysis, spanning from 2015 to 2020, includes both testing and validation phases. In the first wet season, TCCIP observations identify two significant peaks in mean rainfall for 2017 and 2019, which are not detected by the ERA5 reanalysis. This lack of detection in ERA5 is mirrored in the BCSD method, displaying a similar downward trend in both mean rainfall and the RX1day index (Fig.8a, 8b). However, when ERA5 does capture variations in the RX10mm and CDD indices, BCSD shows interannual variation alignment with TCCIP data (Fig.8c, 8d). Conversely, the EDA model more faithfully represents the interannual variability observed in the TCCIP dataset, covering both mean rainfall and extreme rainfall indices (Fig.8d). During the second wet season, characterized by typhoon-induced rainfall, ERA5 more accurately reflects the interannual changes in island-wide mean rainfall, resulting in comparable performance between EDA and BCSD (Fig.8a, 8b, 8c). Notably, the EDA model is better in portraying days of CDD more accurately than BCSD, which tends to underestimate the total count of CDD days significantly (Fig.8d). This superior performance of the EDA model is corroborated by correlation and RMSE metrics presented in Table 4 and aligns with rainfall statistics depicted in Figure 7. In essence, the EDA model provides a more precise depiction of interannual rainfall variations and extreme indices, particularly in correcting the misrepresented variability during the first wet Meiyu season by ERA5. It matches the BCSD model in capturing rainfall extremes and offers a more accurate distribution of CDD, thereby enhancing the model's ability to predict a wide range of rainfall and drought patterns accurately.

**3.5 Extreme Rainfall Event Cases during 2018-2020**

We selected three extreme rainfall events characterized by the highest island-wide daily rainfall occurrences, to evaluate the EDA model's ability to capture the extreme rainfall events. Figure 9 illustrates these rainfall events during the test period from 2018 to 2020. The first event, on May 22, 2020, saw intense rainfall up to 500 mm/day on the southwest side of Taiwan (Fig.9a). This event resulted from multi-scaled interactions involving a strong south-westerly monsoon flow, a southwest vortex, and a potent Meiyu front, which collectively triggered heavy rainfall on the windward slope of the Central Mountain (Chien and Chiu, 2024, 2023). While the ERA5 reanalysis depicted a relatively uniform rainfall distribution across Taiwan with a magnitude of only 100 mm/day (Fig.9b), both BCSD and EDA accurately identified the rainfall peak at the southern tip of the Xue Mountains, with EDA pinpointing the maximum around 22.5˚N but overestimating the intensity to 600 mm/day compared to the observed 500 mm/day (Fig.9c, 9d).

The second significant rainfall event occurred on August 23 and 24, 2018, linked to a tropical depression, as depicted in Figure 9a. This system made landfall in southern Taiwan on August 23 and proceeded to the Taiwan Strait by August 24; a movement observed in the ERA5 reanalysis surface circulations (Fig.9b). The depression's interaction with the existing strong south-westerly flow resulted in enhanced moisture transport into southwestern Taiwan, focusing heavy rainfall on the





windward slopes of Ali Mountain and the Central Mountain, where rainfall peaks reached up to 300 mm/day (Fig.9a; National
Science and Technology Center for Disaster Reduction, 2019). A key difference in the rainfall downscaled by the two models
is evident in BCSD's adherence to the coarse resolution of ERA5, which fails to capture the spatial variability characteristic of
extreme rainfall events (Figs.9b, 9c). Notably, BCSD mimics ERA5 reanalysis by generating increased rainfall over central
Taiwan (~24°N) on both days and weaker rainfall on August 24 over southwestern Taiwan (Fig.9c). In contrast, the EDA
model more accurately delineates the rain-affected regions in alignment with the topography for both days, offering a clearer
north-south differentiation between dry and wet areas.

The third event, occurring on August 24, 2019, involved the severe tropical storm Bailu making a brief landfall at the

southern tip of Taiwan before moving towards the Taiwan Strait. This event resulted in significant rainfall across the eastern
part of the Central Mountain as Bailu (2019) approached (Fig. 9a). ERA5 reanalysis captured the rainfall on the eastern side
of the Central Mountain as the storm neared but tended to overestimate rainfall on the western slope of the Central Mountain
and Ali Mountain, likely due to its coarse resolution (Fig.9b). This overestimation by ERA5 was similarly reflected in the
BCSD model, which inaccurately extended rainfall coverage too far westward, encroaching into the southern part of the Central
Mountain area (Fig.9c). Conversely, the EDA model delineated the precise boundaries of the windward rainfall events, offering
a more accurate representation of the interactions between the storm's dynamics and Taiwan's topography (Fig.9d). However,
upon closer examination, the EDA model's rainfall pattern appears smoother compared to the TCCIP rainfall, indicating a
limitation in capturing the localized, cell-like structures of rainfall, particularly on the northeastern part of Taiwan (Fig.9d).

In summary, the EDA model proves to be particularly adept at replicating extreme rainfall events that arise from the

complex interactions between landscape and synoptic weather circulations. This capability is evident not only across average
seasonal scales but also in accurately depicting the nuances of extreme rainfall events, underscoring its advanced performance
and utility in forecasting and analysing rainfall patterns influenced by topographical features.

### 3.6  Sensitivity experiments with hyperparameters and training/validation periods

### 3.6.1 Impacts of using Surface Winds as Input

To assess the significance of incorporating surface wind data into our model, we conducted an additional experiment

using only rainfall data as input, referred to as EDA_PR (Table 5). This experiment aimed to evaluate the model's performance
in accurately capturing seasonal mean rainfall, particularly in areas known for rainfall hotspots arising from the interaction
between monsoonal winds and topography. Figure 10 illustrates the discrepancies between EDA_PR and TCCIP observations
regarding mean rainfall and climate extreme indices. Figure 10a reveals that mean rainfall is significantly underestimated by
EDA_PR when relying solely on ERA5 rainfall data as input, especially in windward slope areas associated with rainfall
maxima. This outcome indicates that the model's ability to replicate accurate rainfall magnitudes heavily relies on surface wind
data. Further analysis of extreme indices with EDA_PR, as presented in Figures 10b and 10c for RX10mm and CDD,





respectively, aligns with the observations on mean rainfall. The spatial frequency of days experiencing RX10mm is notably
reduced, by up to 10 days, particularly during the first and second wet seasons (Fig.10b). This reduction highlights the impact
of surface wind information on the occurrence of intense rainfall events. Conversely, predictions of CDD days with the
EDA_PR model exhibit varied adjustments when excluding wind data (Fig.10c). During spring, the second wet season, and
winter, CDD is generally underestimated across the island. However, for the first wet season and fall, EDA_PR overestimates
CDD in coastal plains but underestimates it in mountainous regions during fall. Tables 3 and 4 provide a summary of the
CORR and RMSE for EDA_PR, revealing a consistent decline in model performance across most seasons and extreme indices
when compared to the full EDA model. This decline underscores the critical role of surface wind data in enhancing the
proposed model's predictive accuracy and its ability to capture the nuances of rainfall patterns influenced by local surface
circulations.

**3.6.2 Differences between Validation and Test Periods**

In this section, we delve into the disparities between the test and validation periods, serving as a basis for validation and
illuminating potential challenges in rainfall data sampling. Figure S1 delineates the seasonal mean rainfall distribution during
the validation period, highlighting notable variances across seasons, particularly in the second wet season where typhoon-
induced rainfall significantly influences the mean seasonal rainfall and rainfall events. From the data science perspective, the
predictability of typhoon rainfall is heavily contingent on its trajectories, suggesting that the typhoon rainfall samples in the
training dataset may not adequately represent the characteristics of typhoon rainfall in the test and validation periods (cf. Fig.3
and Fig.S1). This discrepancy poses a greater challenge for predictions based on historical rainfall data along with predicting
future changes of typhoon seasons on local scales, adding more uncertainties when estimating future changes of typhoon
rainfall on local communities.
Table 3 and Table 4 further illustrate the fluctuating performance in terms of CORR and RMSE between the test and
validation periods, especially concerning extreme indices. This observation aligns with findings from many previous
downscaling studies, which underscore the necessity of incorporating stochastic elements into downscaling methods to account
for the uncertainties associated with the randomness of rainfall extremes (e.g. Palmer, 2022). Echoing the suggestions of
numerous studies, adopting reinforcement neural networks, such as generative adversarial networks, could offer a promising
solution for capturing the small-scale variability inherent in rainfall extreme (Harris et al., 2022; Price and Rasp, 2022; Oyama
et al., 2023). These advanced modelling techniques may provide a more nuanced understanding and prediction capability for
the complex dynamics of extreme rainfall events.

**4 Discussion and Summary**

Our study underscores the potential of the proposed DNN model with multi-head attention mechanism, the EDA, to
enhance univariate rainfall downscaling, specifically in accurately transitioning coarse-resolution rainfall data to a finer, local-



scale resolution by incorporating auxiliary topographical and surface circulation data. By utilizing the ERA5 reanalysis as input, our primary focus was on mitigating the biases associated with orographic rainfall —a common challenge arising from the limited resolution and parameterization of global models. Taiwan, with its extensive network of rainfall observations and a diverse climate characterized by significant orographic influence on precipitation patterns, offered an ideal setting for this study. This choice allowed for a thorough assessment of the EDA model's capacity to detect and correct biases and variability in daily rainfall data, showcasing their potential in enhancing the accuracy of downscaling methodologies in regions with complex climatic and topographical dynamics.

Our comprehensive analysis, encompassing evaluations of seasonal rainfall, ETCCDI extreme indices, and their internal variations, underscores the EDA model's proficiency in correcting rainfall biases from the ERA5 reanalysis across various seasons. Its performance, in terms of correlation (CORR) and root mean square error (RMSE) across seasonal rainfall and climate extreme index patterns, is on par with that of the BCSD method, especially in seasonal mean pattern. Upon closer examination, however, the EDA model exhibits superior capabilities in amending the overly frequent occurrences of weak rainfall and in accurately addressing instances of heavy rainfall identified in the TCCIP observations, outperforming the BCSD method in these respects. This enhanced performance is particularly notable during the first and second wet seasons in Taiwan, characterized by extreme rainfall events. The EDA model's improved adjustments are evident in climate extreme indices that capture both ends of the rainfall spectrum, such as CDD and rainfall exceeding 10 mm (RX10mm), thanks to the incorporation of surface circulation data. Additionally, the EDA model surpasses the BCSD method in predicting interannual variations of seasonal rainfall and climate extremes, areas where the BCSD method struggles, especially when the parent ERA5 data inaccurately represents rainfall variability.

Our research sets itself apart by deploying a DNN model tailored to Taiwan's distinct climate characteristics, marked by its intricate weather systems and pronounced topographical impact on rainfall distribution. This approach proves effective in addressing the limitations associated with QM-type methods, such as the artificial adjustment of rainfall extremes and inaccuracies in rainfall location due to complex terrain (Maraun and Widmann, 2018). Our application of a DNN model for downscaling not only validates the effectiveness of DL models in refining downscaling methods for climate purposes but also underscores their flexibility to incorporate auxiliary data. Our analysis of rainfall pattern and statistics shows that this inclusion is crucial for representing the intricate rainfall pattern determined by dynamics between topography and atmospheric circulations. Parallel to our focus on climate downscaling, Hsu et al. (2024) found that a CNN-based model excels in amending rainfall patterns for weather forecast datasets on an hourly basis across Taiwan. Moreover, Mardani et al. (2024) showcased the effectiveness of combining U-net with a diffusion model for enhancing the downscaling of kilometer-scale surface variables in Taiwan, mimicking the output of data-assimilated regional climate models. The findings from our study and their studies open new avenues for advancing downscaling techniques, especially in areas like Taiwan where precise rainfall forecasting is essential for managing water resources and preparing for emergencies.





Our future research works are oriented along two primary trajectories. Firstly, we plan to leverage our comprehensive
understanding of Taiwan's rainy seasons, derived from detailed observational studies, to refine our identification of rainfall
characteristics that are most precisely captured by the EDA model. Acknowledging the room for enhancement in terms of
model explainability, we are set to investigate novel methodologies to unravel the DNN model's learning mechanisms. This
initiative aims to elevate the transparency of the model, illuminating the underpinnings of its predictions and enriching our
insight into the model's intrinsic biases. Secondly, having established the EDA model's proficiency in translating coarse-
resolution reanalysis biases into accurate local-scale rainfall predictions, our next objective is to broaden our analysis to
encompass a wider range of realistic applications that involve significant large-scale biases. Our approach involves diversifying
beyond univariate rainfall forecasts to include additional climate variables, thereby enriching the model's downscaling
capabilities. This approach will be informed by existing research that has successfully employed an array of both free-
tropospheric and surface variables as predictors for regional downscaling (Baño-Medina et al., 2021, 2020; Doury et al., 2023).
We intend to initiate this expansion by applying selected CMIP6 models for climate downscaling, aiming to generate precise
local-scale climate projections for Taiwan, by harnessing data from East Asia or potentially global reanalysis. Through these
focused lines of inquiry, we anticipate making substantial contributions to the precision of climate downscaling techniques
and the broader understanding of regional climate dynamics.

**Code and data availability**

The exact version of the DNN downscaling model, EDA, used to produce the results used in this paper, and scripts to run the
model are archived on Zenodo (Chiang, 2024a), as are output data to produce the plots for all the simulations presented in this
paper (Chiang, 2024b). The input data, TCCIP daily rainfall dataset, can be downloaded from the project website
(https://tccip.ncdr.nat.gov.tw/index_eng.aspx), and the rainfall and surface variables from the ERA5 reanalysis can be
downloaded from the Copernicus Climate Change Service (Hersbach et al., 2023).

**Competing Interests**

The authors declare that they have no conflict of interest.

**Author contribution**

YCW, KCW, and WLT has conceptualized the idea and experiment designs, and supervised for this project. CHC developed
the model code, fine-tuned the models, and performed the simulations. CJS conducted data analysis and model evaluations.
CTC has provided important suggestions on research context. HCL has prepared and curated all dataset. YCW organized the
research results, and prepared the manuscript with contributions from all co-authors.




**Acknowledgments**

This work was supported by the Taiwan National Council of Science and Technology under grant numbers NSTC 112-2923-M-001-003. We are also grateful to the National Center for High-Performance Computing of Taiwan for providing the facilities for model training, testing, and validation.

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






| Name | resolution | Time period | Variables |
|---|---|---|---|
| **ERA5 reanalysis** | 0.25˚x0.25˚ global | 1960-2020 | 10m U, V (m/s), rainfall (mm/day) |
| **TCCIP daily rainfall** | 0.05˚x0.05˚ Taiwan land | 1960-2020 | Rainfall (mm/day) |
| **Topography data of Taiwan** | 0.01˚x0.01˚ Taiwan land | static | Altitude (meter) |

Table 1: Data range for training and ground truth dataset over Taiwan.

| Name | Time period |
|---|---|
| **Spring** | Feb 15th-May 15th |
| Summer 1st wet season | May 16th-July 24th |
| **Summer 2nd wet season** | July 25th-September 27th |
| **Autumn** | September 28th -December 1st |
| **Winter** | December 2nd -February 14th |

Table 2: Definition of 5 raining seasons of Taiwan based on previous climatological studies (Chou et al., 2009).

| | CORR (RMSE) | **Spring** test | validation | **1st-wet** test | validation | **2nd-wet** test | validation | **Fall** test | validation | **Winter** test | validation |
|---|---|---|---|---|---|---|---|---|---|---|---|
| meanR | bcsd | 0.93(0.74) | 0.95(0.67) | 0.93(2.97) | 0.92(2.24) | 0.94(1.75) | 0.93(1.59) | 0.97(1.50) | 0.94(1.61) | 0.95(1.42) | 0.97(0.74) |
| | EDA | 0.91(0.76) | 0.94(0.77) | 0.94(1.79) | 0.93(1.77) | 0.94(2.05) | 0.89(2.12) | 0.90(1.91) | 0.91(1.78) | 0.87(1.56) | 0.89(0.93) |
| | EDA_PR | 0.87(1.25) | 0.93(1.49) | 0.91(4.78) | 0.90(5.09) | 0.91(4.30) | 0.81(3.90) | 0.94(2.73) | 0.87(2.99) | 0.91(2.24) | 0.93(1.44) |

Table 3: Performance metrics for mean rainfall for 5 rainy seasons in Taiwan.





| | CORR (RMSE) | Spring | | 1st-wet | | 2nd-wet | | Fall | | Winter | |
|---|---|---|---|---|---|---|---|---|---|---|---|
| | | test | validation | test | validation | test | validation | test | validation | test | validation |
| RX10mm | bcsd | 0.87(1.10) | 0.91(1.12) | 0.85(4.61) | 0.90(2.38) | 0.90(2.39) | 0.82(2.12) | 0.97(2.56) | 0.96(1.93) | 0.94(2.54) | 0.93(2.21) |
| | EDA | 0.85(0.83) | 0.93(0.86) | 0.93(2.53) | 0.90(2.64) | 0.92(2.62) | 0.87(2.19) | 0.93(2.33) | 0.91(1.92) | 0.90(2.23) | 0.88(1.48) |
| | EDA_PR | 0.80(1.40) | 0.90(1.68) | 0.89(7.49) | 0.81(6.98) | 0.86(6.46) | 0.77(4.47) | 0.94(3.91) | 0.93(2.97) | 0.91(3.47) | 0.85(2.83) |
| CDD | bcsd | 0.72(4.37) | 0.85(4.37) | 0.56(5.70) | 0.57(5.70) | 0.68(5.42) | 0.67(6.07) | 0.88(8.73) | 0.88(5.55) | 0.79(6.61) | 0.78(7.27) |
| | EDA | 0.77(2.67) | 0.88(2.90) | 0.57(4.33) | 0.71(4.15) | 0.76(4.20) | 0.60(3.83) | 0.92(6.07) | 0.88(4.10) | 0.83(4.96) | 0.81(5.20) |
| | EDA_PR | 0.65(3.67) | 0.85(3.29) | 0.60(5.05) | 0.61(5.06) | 0.61(4.62) | 0.63(3.81) | 0.86(7.55) | 0.83(4.85) | 0.75(5.88) | 0.76(5.83) |
| RX1day | bcsd | 0.83(11.76) | 0.91(8.73) | 0.87(39.59) | 0.70(60.16) | 0.85(37.54) | 0.90(42.64) | 0.89(20.17) | 0.86(28.85) | 0.85(19.04) | 0.82(12.61) |
| | EDA | 0.90(6.78) | 0.92(6.23) | 0.84(37.87) | 0.84(38.99) | 0.87(36.26) | 0.83(57.23) | 0.88(19.52) | 0.86(27.97) | 0.76(12.19) | 0.82(6.63) |
| | EDA_PR | 0.77(15.17) | 0.86(14.17) | 0.75(61.19) | 0.67(86.92) | 0.78(66.28) | 0.74(83.55) | 0.82(26.83) | 0.66(49.52) | 0.70(20.75) | 0.77(14.54) |

Table 4: Performance metrics for selected extreme indices for 5 rainy seasons in Taiwan. All the metrics are compared with the 5-km grids of the TCCIP observational rainfall.

| Model names | inputs | Output |
|---|---|---|
| EDA_PR | ERA5-rainfall | rainfall |
| EDA | ERA5-rainfall, 10m winds | rainfall |

Table 5: List of EDA models trained with rainfall-only and both rainfall and surface winds.

Figure 1: (a) Taiwan's topography and six observational stations, highlighting Xue Mountain (XM) and Central Mountain (CM). (b) the annual rainfall cycle using the TCCIP dataset, comparing the climatological mean (black), test period (red), and validation period (purple). (c) mean rainfall (mm/day) from TCCIP and mean near-surface streamline of winds from ERA5 across Taiwan's five seasons during the test period. Elevation contours at 1000 meters and 2000 meters are represented by thick black lines, and the boundaries of each county in Taiwan are depicted with fine black lines.

698

699





Figure 2: Model architecture of the Encoder-Decoder with Multi-Head Attention for Auxiliary Channels (EDA) Model. 'N' represents an adjustable parameter that dictates the repetition frequency of model components. Data dimensions at each layer are annotated, with 'b' indicating the batch size utilized.



Figure 3: Mean rainfall distribution of 5 rainy seasons defined in Table 1 with units of mm/day during test period (2017/12/13-2020/12/31). (a) ERA5 reanalysis, (b) BCSD downscaled rainfall from ERA5, and (c) EDA downscaled rainfall from ERA5. Elevation contours at 1000 meters and 2000 meters are represented by thick black lines, and the boundaries of each county in Taiwan are depicted with fine black lines.







Figure 4: (a) Spatial distribution of RX1day index from TCCIP observations and downscaled bias in (b) BCSD and (c) EDA
model during test period (2017/12/13-2020/12/31). Elevation contours at 1000 meters and 2000 meters are represented by
thick black lines, and the boundaries of each county in Taiwan are depicted with fine black lines.



(a)

(b)

(c)

Figure 5: (a) Spatial distribution of RX10mm index from TCCIP observations and downscaled bias in (b) BCSD and (c) EDA model during test period (2017/12/13-2020/12/31). Elevation contours at 1000 meters and 2000 meters are represented by thick black lines, and the boundaries of each county in Taiwan are depicted with fine black lines.





Figure 6: Spatial distribution of CDD index from (a) TCCIP observations and downscaled bias with (b) BCSD and (c) EDA model during test period (2017/12/13-2020/12/31). Elevation contours at 1000 meters and 2000 meters are represented by thick black lines, and the boundaries of each county in Taiwan are depicted with fine black lines.

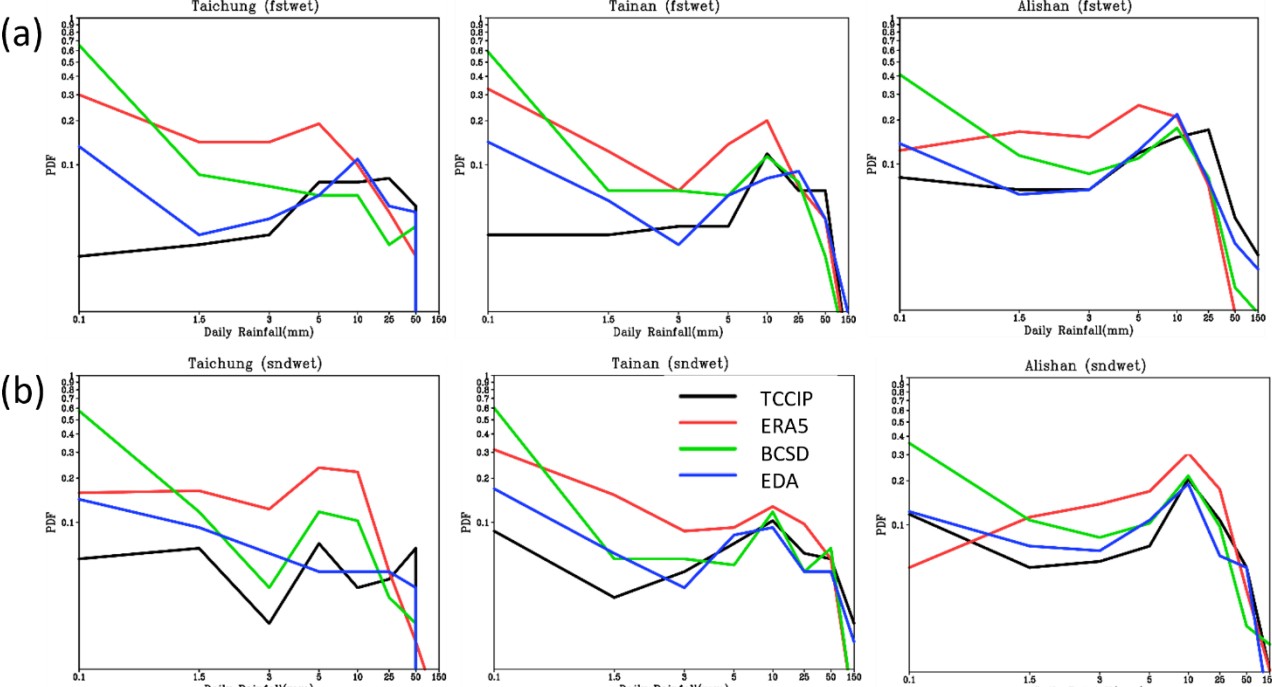

Figure 7: Rainfall distribution for selected Central Weather Administration (CWA) observational stations during (a) the first wet season and (b) the second wet season. For the summer seasons, stations located on the western plains and mountains, including Tainan Station, Taichung Station, and Alishan Station, are featured. The arrangement of the stations in the figure follows a north-to-south order, based on the latitudinal positions of their locations, with stations positioned from top to bottom accordingly (Figure 1a).




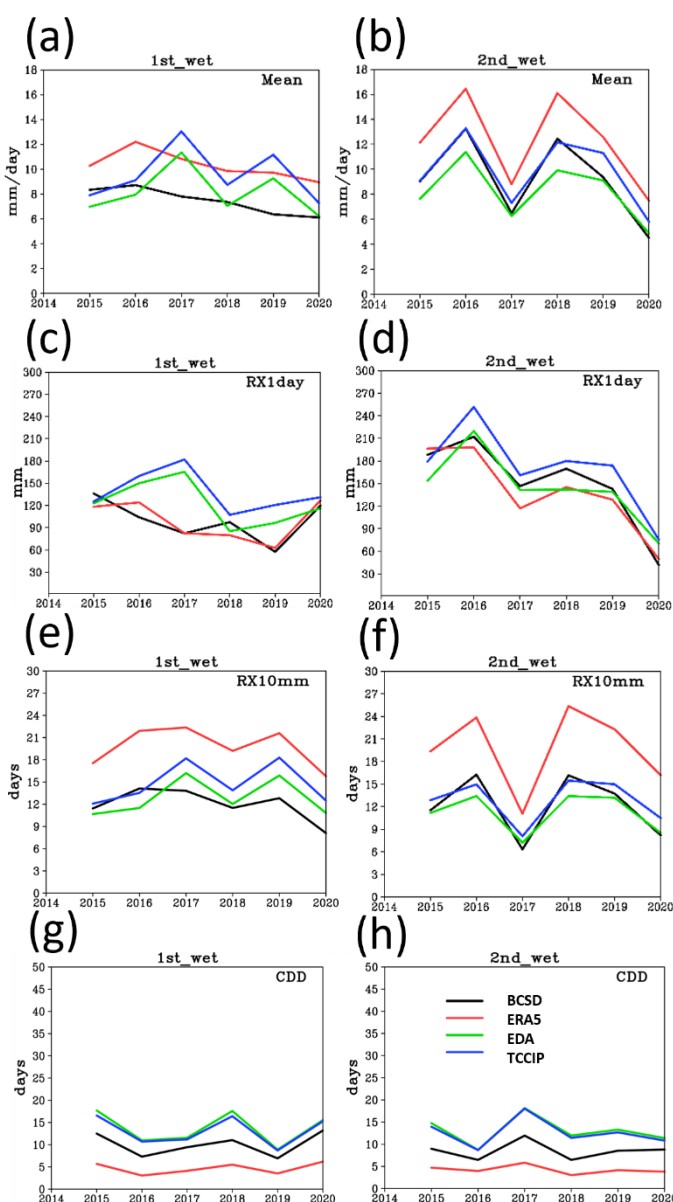

Figure 8: Interannual variation of island-wide mean rainfall and selected extreme indices based on TCCIP observations (blue), ERA5 (red), and EDA (green), and BCSD (black) for the (a,b) mean, (c,d) RX1day, (e,f) RX10mm, and (g,h) CDD indices. From left to right, columns indicate the 1st wet season and the 2nd wet seasons during validation and test period.







(a)

(b)

(c)





(d)

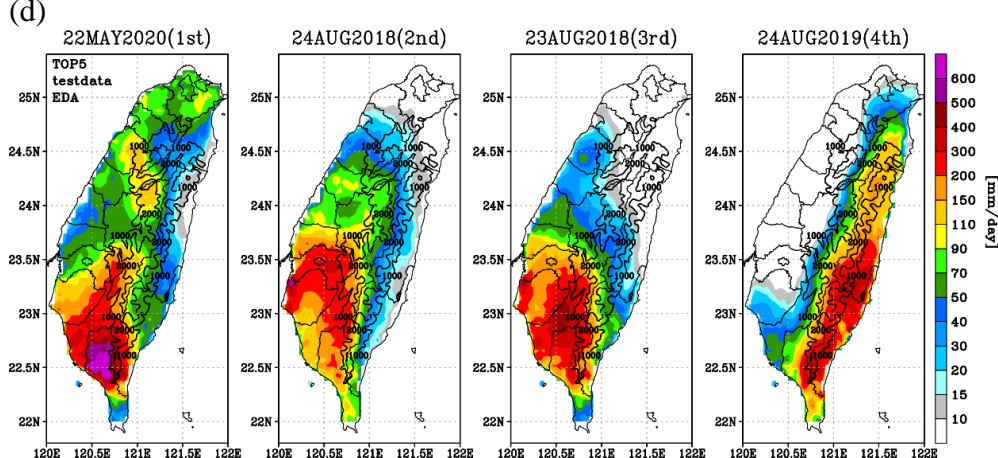

Figure 9: Daily rainfall distribution of the 3 extreme rainfall events based on TCCIP daily rainfall during test period (**2017/12/13-2020/12/31**). Here we choose 4 days from the rainfall ranking of the highest 5 days of island-wide rainfall average during this period. From left to right, they are May 22, 2020, August 24, 2018, August 23, 2018, and August 24, 2019. From the synoptic analysis of these rainfall days shows the May 22, 2020 event is related to a Meiyu frontal system, the event of August 23-24, 2018 is related to a tropical low-pressure system, and August 24, 2019 is related to Typhoon Bailu. (a) for TCCIP data set, (b) for ERA5 reanalysis, (c) BCSD downscaling method, and (d) for EDA downscaling model. Elevation contours at 1000 meters and 2000 meters are represented by thick black lines, and the boundaries of each county in Taiwan are depicted with fine black lines.




(a)

(b)

(c)

Figure 10: Spatial distribution of downscaled rainfall based on EDA model structure but with only rainfall as the input for 5 rainy seasons in Taiwan. (a) mean rainfall, (b) RX10mm, and (c) CDD difference from TCCIP during test period (**2017/12/13-2020/12/31**).