# Peer review of "Using Multi-Head Attention Deep Neural Network for Bias Correction and"

_EGUsphere, 2024_

## Referee Comment (RC1)

**Using Multi-Head Attention Deep Neural Network for Bias Correction and Downscaling for Daily Rainfall Pattern of a Subtropical Island**

**General comments**

In this work, authors propose an attention-based deep learning model for the downscaling and bias correction of precipitation over Taiwan. By relying on precipitation and 10-m wind at a coarse scale (25 km), the model is able to generate the downscaled field at a regional scale (5 km). To achieve this, they use a reanalysis dataset as the input (predictor) and an observational dataset as the output (predictand). They also integrate orographic data into the model to better represent local precipitation affected by orographic phenomena. By comparing the proposed model against a baseline bias-correction algorithm, they demonstrate the improvement of the proposed model across a wide set of metrics, including those characterizing the mean, precipitation extremes, and interannual variability, as well as specific extreme precipitation events. As a result, the authors plan to continue working on applying this model to climate models to generate regional-scale projections for future scenarios.

First of all, I recommend that the authors review the use of the English in the manuscript, particularly with respect to some sentence structures and inconsistent verbal tenses. Also, I believe that the organization of the manuscript and presentation of the data and results need some improvement. I leave some specific comments regarding this in the *Specific Comments* section.

From my perspective, there are some misconceptions regarding important concepts in the field of deep learning for statistical downscaling. For instance, in the introduction (L36-49), authors define statistical downscaling as follows:

*"Statistical downscaling, conversely, constructs empirical relationships between coarse-resolution variables from GCMs and local-scale surface variables"*

However, as defined in [1]:

*"In statistical downscaling, empirical links between the large-scale and local-scale climate are identified and applied to climate model output."*

Whether these links are constructed using simulations/models (e.g., GCMs) or observations/reanalysis as input characterizes the Model Output Statistics (MOS) and Perfect Prognosis (PP) approaches, respectively. Thus, the authors are wrongly defining statistical downscaling as MOS, since the latter is a subset of the former.

In the next paragraph, authors define Super Resolution (SR) techniques. As they correctly mention, these techniques have gained popularity due to advances in the deep learning field. However, the main drawback of SR models is that they generally rely on the upscaled surface variable or an equivalent from a different observational/reanalysis dataset (e.g., ERA5). One could relate this approach to Perfect Prognosis (PP); however, in PP, models are not built with surface variables as predictors (among other assumptions), as these are not well reproduced by GCMs (this would lead to biased projections when downscaling the GCM). This is especially relevant for variables with a heavy local-influence such as precipitation, which may substantially differ between observations and GCMs. Instead, PP relies on large-scale synoptic variables, which are properly simulated by GCMs.

SR techniques are also inappropriate for Model Output Statistics (MOS) in the climate context, as these techniques assume a day-to-day correspondence between the simulated data (GCM in this case) and observations, which is not fulfilled in this context. This is why, in the climate context, MOS is performed distributionally (e.g., the BCSD technique authors use as baseline).

These considerations align with the downscaling model proposed in this paper. In this work, authors downscale precipitation (5 km) by relying on precipitation and 10-m wind data from ERA5 (25 km). Since this framework does not align neither PP or MOS assumptions, I recommend removing all mentions of applications of the proposed deep learning model to GCMs and instead framing it within the context of SR.

I would also like to highlight that the deep learning model proposed in this paper is based on the one developed in [2], with the addition of the attention mechanism and the 10-m wind covariate. However, the experimental framework of these two works is very similar. In Section 3.6.1, the effect of including the 10-m wind variable is addressed. However, to justify the use of attention-based layers, the proposed model should be compared to that of [2], or at least to some other CNN model previously discussed in the literature (e.g., [3]).

**Specific comments**

**L23***"[...] between synoptic and local circulations influenced by topography."***:** The proposed model is fed precipitation and 10-m wind as predictors, which are considered surface variables rather than synoptic variables.

**L36-49:** I suggest rewriting this paragraph following my general comment on the definition of statistical downscaling.

**L43** *"[...] the Model Output Statistics (MOS) [...] requiring no prior knowledge for the of predictors or regions."*: I believe that such a strong statement requires a reference.

**L64-69:** I would recommend trying to better categorize these papers based on my comments on the definition of statistical downscaling in the general comments section.

**L91-93/153-155** *"we have refined the training and validation procedures by opting for partitioning based on consecutive years"*: This practice has been already used in numerous downscaling works, such as [4, 5], among others. I believe these should be mentioned.

**L101**: Are environmental *representations in GCMs* the same as the synoptic scale?

**L101-102** *"Our methodology is tailored for future climate downscaling rather than nowcasting"*: As I mention in the general comments, I believe authors need to reconsider the scope of this work regarding future applications.

**L120-124:** If, as authors mention, they want to reproduce the interplay between synoptic weather patterns and topography you need to rely on synoptic variables, but the developed model relies on surface ones.

**L149-151:** What is the purpose of the validation set in this study? Generally, this set is used to find the best set of hyperparameters, performing the final evaluation in the test set. May it be used here for the early-stopping process?

**L157-158** *"These steps encompass a log1p transformation to adjust for the skewness in the distribution of the data values, particularly beneficial for precipitation data":* Is this transformation applied to all variables or just to the precipitation? It is not clear.

**L158-159** *"Moreover, we normalize various data variables to ensure consistency across the dataset: precipitation, temperature, and humidity data [...]":* In L124 you say: *"To capture these complex interactions, our model, EDA, incorporates ERA5-derived daily aggregated rainfall and 10-meter height wind data as inputs".* What variables did you really use as predictors? There are some contradictions in the manuscript.

**L186-187** *"[...] As for the downscaling process, the intermediate outputs from the encoder are transitioned to the decoder, [...]":* I do not understand this. Following Figure 2, it seems that the intermediate outputs of the encoder are not transitioned to the decoder, instead the final encoder's output is passed to the decoder. I believe this needs further clarification.

**L192-197**: I understand that the use of the pixel-shuffle layer for upscaling is inspired by the Enhanced Super-Resolution CNN, but what about the Super-Resolution Using Deep Convolutional Networks you mention in the paragraph before? Is any element of this model used in the model proposed in this work?

**L202-203**: Regarding the WMSE loss function, is this the first time such a metric is used or did you draw inspiration from some other work? Also, I think it is necessary to specify what value of γ do you set. If you choose γ=0 you get the standard MSE, otherwise if γ=1 you get the MSE weighted by the true precipitation value. The latter can be problematic for non-precipitation cases, as these would not contribute to the loss function. This is problematic both for the raw precipitation value as for the lop1p-transformed (non-precipitation would still correspond to 0).

**L205-208**: I believe readers should be provided with more details regarding the pre-training phase of the encoder. If I understand correctly, the encoder is initially pre-trained on low-resolution data. However, it is not explicitly defined what this data corresponds to. Is it a low-resolution version of the target data? Additionally, it's unclear if the same loss function is used during this pre-training phase.

**L206-207** *"[...] the encoder is frozen, and the encoder is trained on high-resolution data, [...]"*: I guess this is a typo, as when you freeze the encoder you train the decoder.

**Figure2:** This figure indicates that the input data have a channel size of 4, which does not align with the variables introduced in Section 2.1 (see my L158-159 correction).

**L230**: I couldn't access Lin et al., 2023, the paper in which the baseline method is based on.

**L257:** What set of additional climate indices are you referring to?

**L290** *"with a bias residual of less than 2 mm/day (Fig. 3b, 3c)"*: You mention the bias residual, but there are no biases plotted for the mean rainfall. I recommend either adding an additional row to this figure displaying the climatology of the observational dataset or following the format of other figures (e.g., Fig 4, Fig 5, Fig 6) and including the bias in the plot.

**Caption of Figure 4, 5, 6, 9, 10**: Why does the test period start on such a specific day (2017/12/13)? Is this a typo, or does the test period actually begin on this specific day?

**L323** *"Figure 5 displays the spatial distribution of days experiencing rainfall exceeding 10 mm (RX10m) [..]"*: Does RX10m represent the number of days or the

amount of precipitation for days with 10m? It is not clear in the text, as in L255 you say *annual account* but then in Figure 5 the colorbar has the label *mm/day*. This needs further clarification.

**Figure 6**: The label of the colorbar for the biases should be days instead of mm/day .

**L340-341** *"During the summer's wet seasons, BCSD consistently overestimates CDD throughout Taiwan"*: If I'm not mistaken, BCSD underestimates the CDD instead of overestimating it.

**L341-342** *"In fall, BCSD's predictions overestimate CDD in northeastern Taiwan and similarly overestimate CDD in western Taiwan"*: In northeastern Taiwan, BCSD underestimates CDD instead of overestimating it. I suggest reviewing this paragraph thoroughly, as there may be some misunderstandings regarding the under/overestimation of the CDD metric.

**L344-345** *"[EDA] tends to underestimate CDD during fall, indicating a nuanced yet imperfect prediction capability for dry periods throughout the seasons."*: Could this also be attributed to the WMSE loss function? Without specific details about the chosen γ in the manuscript, it's uncertain. However, a high value of this hyperparameter might cause the model to overestimate the precipitation amount, consequently resulting in an underestimation of the CDD.

**L347-349** *"The challenge in accurately modelling this season may stem from the substantial contribution of typhoon-related rainfall, which due to its somewhat stochastic nature compared with other seasons, complicates the precise prediction of rainfall distribution"*: Some of the differences shown in Table 4 are quite small, which makes drawing conclusions difficult. Have you considered, instead of presenting results for a single model, training the model multiple times and reporting the mean metric across these runs along with a measure of variability? This approach would facilitate the assessment of whether the EDA model differs significantly from the BCSD.

**Section 3.3:** Why is the zero (no precipitation) not represented in Figure 7? MSE-based deep learning models often tend to fit the mean, which can be problematic for a variable like precipitation. Including visualization of the 0 mm category would be beneficial. Additionally, have you considered using other visualization methods? For example, a histogram with the y-axis in logarithmic scale could provide better visualization of extreme values.

**L378** *"Our analysis, spanning from 2015 to 2020, includes both testing and validation phases"*: I believe that using the validation period to assess the final accuracy of a deep learning model may lead to a biased assessment of its accuracy, as this period

is typically utilized to search for the optimal configuration. If the validation period has not been used to tune the model, then it should be considered part of the test period.

**L436-437** *"Figure 10a reveals that mean rainfall is significantly underestimated by EDA_PR when relying solely on ERA5 rainfall data as input [...]"*: This is the first time the bias of the mean is shown. In Figure 3, the mean is displayed, but not the bias with respect to the target data. Would it be possible to also show the bias in the mean for the EDA model? This would facilitate comparing EDA_PR with EDA in terms of this metric.

**L464** *"[...] adopting reinforcement neural networks, such as generative adversarial networks, [...]"*: As far as I know, reinforcement and generative learning are two distinct paradigms. Therefore, a generative adversarial network is not a reinforcement neural network.

**Technical corrections**

**L29-30:** I suggest including [6] (Section 1.5.3) as reference for GCMs/ESMs.

**L104-109:** Instead of Sections, the text wrongly refers to these as Sessions.

**L144** *"[...] as outlined by X"*: This appears to be a typo.

**L176:** This should refer to Figure 2, not Figure 3.

**L166** *"[...] an encoder and an encoder"*: This should be *"an encoder and a decoder"*

**L220** *"[...] for its capability to maintain the mean percentile of data distribution efficiently, [...]"*: I believe this sentence could be improved, for instance by simply saying: *"for its capability to reproduce the mean".*

**L241-243**: This paragraph needs some rephrasing for better understanding.

**L253:** SDII stands for Simple Daily Intensity Index, not Simple Precipitation Intensity Index).

**L255**: Following [7], this metric should be denoted as R10mm, not RX10mm.

**L272** *"[...](depicted by red and purple lines in Fig.1b)[...]"*: Are you referring to the test and validation precipitation series? If so, you could simply mention that such extreme precipitation events occur during these periods.

**L276** *"[...](Fig.1b)[...]"*: I believe you are not referring to Fig. 1b here, but rather Fig. 1c.

**L282-286**: You are referencing subplots that are not defined (e.g., Fig 1e or Fig 1f).

**Figure 8**: This figure requires some improvements in terms of visual appearance, such as addressing issues like no data for 2014 being plotted on the x-axis and the presence of empty space in half of the plot. Additionally, it would be beneficial for the reader to use consistent colors to represent the same models across Figures 7 and 8.

**References**:

[1] Maraun, D., & Widmann, M. (2018). Statistical downscaling and bias correction for climate research. Cambridge University Press.

[2] Chiang, C. H., Huang, Z. H., Liu, L., Liang, H. C., Wang, Y. C., Tseng, W. L., ... & Wang, K. C. (2024). Climate Downscaling: A Deep-Learning Based Super-resolution Model of Precipitation Data with Attention Block and Skip Connections. arXiv preprint arXiv:2403.17847.

[3] Vandal, T., Kodra, E., Ganguly, S., Michaelis, A., Nemani, R., & Ganguly, A. R. (2017, August). Deepsd: Generating high resolution climate change projections through single image super-resolution. In Proceedings of the 23rd acm sigkdd international conference on knowledge discovery and data mining (pp. 1663-1672).

[4] Baño-Medina, J., Manzanas, R., & Gutiérrez, J. M. (2020). Configuration and intercomparison of deep learning neural models for statistical downscaling. Geoscientific Model Development, 13(4), 2109-2124.

[5] Harris, L., McRae, A. T., Chantry, M., Dueben, P. D., & Palmer, T. N. (2022). A generative deep learning approach to stochastic downscaling of precipitation forecasts. Journal of Advances in Modeling Earth Systems, 14(10), e2022MS003120.

[6] Chen, D., M. Rojas, B.H. Samset, K. Cobb, A. Diongue Niang, P. Edwards, S. Emori, S.H. Faria, E. Hawkins, P. Hope, P. Huybrechts, M. Meinshausen, S.K. Mustafa, G.-K. Plattner, and A.-M. Tréguier, 2021: Framing, Context, and Methods. InClimate Change 2021: The Physical Science Basis. Contribution of Working Group I to the Sixth Assessment Report of the Intergovernmental Panel on Climate Change[Masson-Delmotte, V., P. Zhai, A. Pirani, S.L. Connors, C. Péan, S. Berger, N. Caud, Y. Chen, L. Goldfarb, M.I. Gomis, M. Huang, K. Leitzell, E. Lonnoy, J.B.R. Matthews, T.K. Maycock, T. Waterfield, O. Yelekçi, R. Yu, and B. Zhou (eds.)]. Cambridge University Press, Cambridge, United Kingdom and New York, NY, USA, pp. 147–286, doi:10.1017/9781009157896.003.

[7] https://etccdi.pacificclimate.org/list_27_indices.shtml

---

## Referee Comment (RC2)

**Review of the manuscript:**

**Using Multi-Head Attention Deep Neural Network for Bias Correction and Downscaling for Daily Rainfall Pattern of a Subtropical Island**

In this paper, the authors proposed a deep learning model called EDA to downscale the daily rainfall precipitation from ERA5 with $0.25^{\circ} \times 0.25^{\circ}$ to 5-km observed rainfall data over Taiwan. The paper can be seen as an improved version of Chiang et al. (2024) where the authors replaced the channel and spatial attentions (Woo et al. 2018) by a self-attention from Vaswani et al. (2017). In addition, 10-m wind components and high-resolution topographic data were used as auxiliary information and the loss function was replaced by a weighted mean squared error (WMSE). The authors show that the model can perform a downscaling and bias correction from ERA5 to match the high-resolution of TCCIP observational data.

**General comments**

First, since the manuscript introduces an improved version of an existing model from Chiang et al. (2024), it should provide ablation studies about the newly introduced model components. In my opinion, this should definitely include ablation studies about the:

- proposed loss function.
- self-attention layers in comparison with the one from Chiang et al. (2024) and with CNN.
- encoder block in comparison with the standard block design from Transformer by Vaswani et al. (2017).
- training procedure with two phases (see lines 205-208).
- sensitivity studies about auxiliary information i.e., topographic data and temperature/humidity.

I also find it confusing to name the attention module *"multi-head Attention layers for auxiliary channels (EDA)"*. As mentioned in lines 170-172: *"…This capability is pivotal for our model, allowing it to simultaneously process the entire dataset and enabling each grid point to evaluate its relationship with all others…"* the layer performs attentions across the spatial domain and not between the channels i.e., the attention matrix will be $126 \times 126$ in this case compared to $4 \times 4$ in case of using 4 auxiliary predictors without increasing the dimensionality (see Fig. 2).

Second, I think the manuscript should indicate clearly what ERA5 variables were used in the study. Throughout the manuscript, it is stated that 10-m wind components were used as auxiliary predictors while line 159 introduces suddenly two more predictors (temperature and humidity). Figure 2 implies also that 4 input variables were used.

Third, it looks like there is a misunderstanding about the applications of this study. The authors wrote in lines 101-102: *"…Our methodology is tailored for future climate*

*downscaling rather than nowcasting…"* and in lines 514-515: *"…We intend to initiate this expansion by applying selected CMIP6 models for climate downscaling, aiming to generate precise local-scale climate projections for Taiwan, by harnessing data from East Asia or potentially global reanalysis…"* Reanalysis data and climate projections are different. I don't think the experimental framework is designed in a way that the model can be used to downscale climate projections since training was done on ERA5 reanalysis.

**Lines 162-163** *"These improvements collectively contribute to an increase in the model's computational efficiency and predictive accuracy":* What do the improvements refer to? Did the manuscript prove that the referred improvements increased computational efficiency and accuracy? If yes, where in the manuscript?

**Lines 205-208** *"…The training regime is executed in a supervised manner, with an initial focus on training the encoder using low-resolution observational data. Subsequent to this phase, the encoder is frozen, and the encoder is trained on high-resolution data, a strategy designed to fine-tune the model's ability to perform accurate downscaling and bias correction…":* Where was it proven in the manuscript that the newly introduced strategy improves the accuracy of downscaling and bias correction?

**Subsection 2.3: Model Structure:** In my opinion and since the manuscript is submitted to (GMD), this section lacks essential implementation details. For instance, what is the number of layers in the encoder (*N* in Fig. 2)? What are the kernel sizes, strides and paddings for CNN? How is the positional embedding implemented? is it learnable or based on a fixed sinusoidal embedding? What kind of activation function is used? How are the topographic data normalized? What kind of normalization layers is used in the model? is it LayerNorm Ba et al. (2016)? What is the drop out ratio? How is the Up-sampling done? What does subpixel mean? Is it interpolation or ConvTranspose? How many attention-heads are being used? How big is the model and how many parameters does it have? In addition, self-attention should be described technically.

**Implementation details:** I found some differences between the submitted code on Zenodo (Chiang, 2024a) and the parameters mentioned in the manuscript. For instance, $\gamma$ is set to zero in the code, early stopping was set to 20 epochs instead of 60 and learning rate was set to 0.00005 instead of 0.0001 in the manuscript.

**Subsection 2.4 Baseline Downscaling Methods:** Since the experimental framework and data are very similar to the ones used in Chiang et al. (2024), I think the deep learning models from Chiang et al. (2024) should be included as baselines.

I would also like to emphasize that the manuscript should acknowledge the limitations i.e., the computational requirements for self-attention grow quadratically with resolution or that the model was trained with a biased corrected Reanalysis (ERA5) as input and not with a climate projection which makes it unapplicable to downscale climate projections.

I also suggest that the authors rearrange the manuscript in a way that they move the figures and tables from the appendix and put them in context within the main body of the manuscript. Furthermore, the manuscript needs to be checked for references to tables and

figures inside the text and the usage of language, especially regarding the conjunctions and connecting words/sentences (see specific and technical comments).

**Specific and technical comments**

**Lines 10-13** *"Abstract. This study investigates the capability of a deep learning approach, employing a multi-head attention mechanism within a deep neural network (DNN) framework, aimed at refining the bias correction and downscaling process for the fifth generation European Centre for Medium-Range Weather Forecasts reanalysis rainfall datasets to provide local-scale daily rainfall data across Taiwan, a mountainous subtropical island..."*: I think this is a very long sentence. Please split the sentences and ideas in the manuscript.

**Line 87:** grammar, mean squared error (MSE).

**Lines 96-97** *"...incorporating surface winds and topography as auxiliary datasets...":* did you only used wind components as predictors?

**Line 140:** What kind of topographic data is used? Slope, DEM, ...?

**Lines 29-30:** Please add references for GCM and ESM.

**Line 53:** Do you mean details?

**Line 54:** What do you mean by *"...on pure-resolution approaches..."*?

**Line 58:** Do you mean downscaling instead of upscaling?

**Lines 59-60** *"...Notably, the integration of skip connections within the encoder-decoder architecture, as seen in the YNet model developed by Liu et al. (2020), is a significant advancement..."*: what is meant here? Skip connection was first introduced in U-Net Ronneberger et al. (2015) to prevent the vanishing gradient and information loss through the bottleneck and to transfer fine details from the encoder levels to the corresponding decoder levels. I don't think adding skip connections is a significant advancement now.

**Fig.1 (c):** It is hard to understand the legend *"meanr testdata obs"*?

**Line 106:** BCSD is first introduced without an explanation.

**Line 124:** What type of wind data did you use  (speed/direction u/v)?

**Line 146:** Equations should be numbered sequentially. Please follow the guidelines from https://www.geoscientific-model-development.net/submission.html#math.

I would also add the local predictors such as topographic data to the equation.

Moreover, $X$ and $Y$ are tensors and should be **X** and **Y. X** should also refer to other predictors from ERA5 such as the 10-m wind not just the rainfall. $f$ is a function representing the neural network.

**Lines 152-153** *"...This separation into distinct sets for testing and validation enables us to more accurately assess the model's predictive uncertainties across varying data regime...":*

This allows to assess the model performance for different time periods. If you want to assess the model performance over different regions you should do a spatial split.

**Line 157:** What is meant by log1p. What does 1 refer to? Is it ln(x+1)?

**Line 159:** What are the wind vector data (speed or direction) and how do you do the normalization for precipitation? Is it ln(precepetation+1)? What is the reason behind normalizing differently for wind and temperature/humidity? Why were the input variables not normalized to have a zero mean and a standard deviation of one?

**Lines 159-160:** What kind of humidity and temperature are being used here?

**Line 161:** Normalization is used to prevent the domination of a subset of the input and to reduce the impact of outliers. In addition, it improves the convergence and keeps the weights in the model within small ranges.

**Line 166:** I think you mean an encoder and a decoder.

**Line 111, subsection 2.1 Data**: What is the spatial resolution of the input images? Is it $14 \times 9$? and in what coordinate system? Why was an image-based approach chosen over the video-based one?

**Lines 168-170** *"…Unlike conventional neural networks that rely on recurrent or convolutional layers, the Transformer architecture is built entirely around attention mechanisms, facilitating direct modelling of dependencies regardless of their distance in the input data…"*: What do you mean by regardless of the distances between the input points? What about the Softmax inside the attention block?

**Line 171:** What is meant here by processing the entire dataset? I think you mean the entire input data points.

**Line 173** *"…This approach not only captures the intricate interdependencies characteristic of climate variables but also introduces flexibility in handling input data of varying sizes…"*: Transformer can't handle varying size due to the positional encoding. Of course, some recent models such as Liu et al. (2022) replaced the relative biases with learnable ones but in principle it can't be stated that transformer can handle a variable input size for spatial data.

**Line 176:** This should be figure 2 instead of 3.

**Line 700, Fig. 2:** Do you mean topographic data and positional encoding? Was the first two fully connected layers used without an activation in between? I would also call the decoder a decoder not a resolver to be consistent across the manuscript. In addition, what is the rationale behind the arrangement of the skip connections and layers in a way that is different to the common building block of the transformer which uses: LayerNorm > Attention > LayerNorm > FullyConnectedLayer?

**Lines 184-188** *"…Decoder part is designed flexibly that one could implement the desired sub-model for combining the intermediate outputs from the encoder with the topography data and performing a one-step upscaling. As for the downscaling process, the intermediate outputs from the encoder are transitioned to the decoder, which are initially reshaped into*

*two-dimensional gridded data before being processed by the decoder…"*: The final output of the encoder not the intermediate output is transferred to the decoder.

**Line 186**: What is the rationale behind using one-step upscaling instead of the common cascading one i.e., like the standard Upsampling in U-Net.

**Lines 198-199** *"…Implementation is carried out within the TensorFlow framework, leveraging its robust capabilities for efficient model training and optimization…"*: I would remove this sentence. Most deep learning frameworks provide efficient training and optimization.

**Line198:** Did you use Keras API? Please make it clear.

**Line 200:** I think 60 epochs for an early stopping is a large parameter.

**Line 202:** Equations should be numbered sequentially. Please follow the guidelines from https://www.geoscientific-model-development.net/submission.html#math.

How is $\gamma$ defined? There is also a typo, the subscripts *I* and *J* for ground truth should be small letters *i* and *j*.

Please either add batch size B or add a sentence to mention that the batch size is omitted for simplicity.

**Line 203:** H and W should be in italic $H$ and $W$.

**Line 205** *"…of the training regime…"*: I would write the training procedure/scheme.

**Line 207:** Do you mean the decoder is trained while the encoder was frozen?

**Line 209:** Please add a reference to Adam optimizer since there are many versions of Adam. Do you use weight decay parameters?

**Line 209** *"…to adjust model parameters effectively during the training process…"*: I would remove this sentence.

**Line 211** *"…a choice that significantly enhances computational efficiency…"*: I would remove this part and just mention the time for training.

**Lines 212-213** *"…This setup ensures that the model is both accurately and efficiently trained to meet the demands of precise climate data downscaling…"*: Please report the inference time.

**Line 257:** Where in the appendix?

**Line 267:** Do you mean in Table 2?

**Line 276:** Do you mean Fig. 1c?

**Lines 274-286:** I couldn't find Fig. 1d - Fig. 1f.

**Line 308:** Where exactly in the appendix?

**Figure 1:** (a) is missing in the top left corner.

**Line 335:** Figure 1a does not have observation.

**Line 452:** I couldn't find Fig. S1.

**Line 464** *"…Echoing the suggestions of numerous studies, adopting reinforcement neural networks, such as generative adversarial networks…"*: Reinforcement learning is not related here. I think you mean generative neural networks such as generative adversarial networks.

**Lines 466-467:** I would mention diffusion models (Rombach et al. 2022) since they are the state-of-the-art in image generation not GANs anymore.

**Line 494:** The term DL is first introduced without an explanation. I think you mean Deep Learning.

**Line 499:** Please add references for U-Net and diffusion models.

**Lines 514-517:** As I mentioned in the major comments, it is questionable how a model trained on reanalysis data such as ERA5 will perform on climate projections.

**Line 518:** It is better to publish the preprocessed data and refer to the raw data.

**Line 527:** Grammar, have.

**Line 528:** What do you mean by performing the simulation? Did you run simulations?

**Line 683:** Table 1 Is not mentioned in the text expect in line 267 where the authors meant Table 2. It is also implied from this table that only wind and precipitation were used from ERA5 while temperature and humidity are not mentioned.

**Line 685, Table 2:** Why are all seasons in bold text except the Summer 1$^{st}$ wet season?

**Lines 687-691, Tables 3 and 4:** To provide statistically significant results, please run the models with 3 different random seeds and report the mean and standard deviations.

**Line 705: figure 3:** Do you mean defined in Table 2? It would be helpful to plot the ground truth from TCCIP in this figure.

**Line 709, figure 9:** I think it is better to put this figure on one page?

**References:**

Ba, J. L., Kiros, J. R., & Hinton, G. E. (2016). Layer normalization. *arXiv preprint arXiv:1607.06450*.

Chiang, C.-H.: EnDeAux_Climate_Downscaling, Zenodo [code], https://doi.org/10.5281/zenodo.10937920, 2024a.

Chiang, C. H., Huang, Z. H., Liu, L., Liang, H. C., Wang, Y. C., Tseng, W. L., ... & Wang, K. C. (2024). Climate Downscaling: A Deep-Learning Based Super-resolution Model of Precipitation Data with Attention Block and Skip Connections. arXiv preprint arXiv:2403.17847.

Liu, Z., Hu, H., Lin, Y., Yao, Z., Xie, Z., Wei, Y., ... & Guo, B. (2022). Swin transformer v2: Scaling up capacity and resolution. In Proceedings of the IEEE/CVF conference on computer vision and pattern recognition (pp. 12009-12019).

Rombach, R., Blattmann, A., Lorenz, D., Esser, P., & Ommer, B. (2022). High-resolution image synthesis with latent diffusion models. In Proceedings of the IEEE/CVF conference on computer vision and pattern recognition (pp. 10684-10695).

Ronneberger, O., Fischer, P., & Brox, T. (2015). U-net: Convolutional networks for biomedical image segmentation. In Medical image computing and computer-assisted intervention–MICCAI 2015: 18th international conference, Munich, Germany, October 5-9, 2015, proceedings, part III 18 (pp. 234-241). Springer International Publishing.

Vaswani, A., Shazeer, N., Parmar, N., Uszkoreit, J., Jones, L., Gomez, A.N., Kaiser, Ł . & Polosukhin, I. (2017). Attention is all you need. Advances in neural information processing systems, 30.

Woo, S., Park, J., Lee, J. Y., & Kweon, I. S. (2018). Cbam: Convolutional block attention module. In Proceedings of the European conference on computer vision (ECCV) (pp. 3-19).

---

## Referee Comment (RC3)

**Using Multi-Head Attention Deep Neural Network for Bias Correction and Downscaling for Daily Rainfall Pattern of a Subtropical Island by Wang et al.**

**Main comments**

In the study "Using Multi-Head Attention Deep Neural Network for Bias Correction and Downscaling for Daily Rainfall Pattern of a Subtropical Island" by Wang et al., a super resolution network is used to bias correct and downscale ERA5 reanalysis rainfall with 25 km horizontal spatial resolution to the observational TCCIP rainfall dataset with 5 km resolution over Taiwan. The performance of the method is extensively evaluated to a combination of quantile mapping and bilinear interpolation. Overall, the deep learning approach shows a promising performance in particular with respect to extreme event statistics.

The authors describe their method as being "tailored for future climate downscaling". In this context, it would be important to test whether the model can also preserve non-stationary trends that are expected in future warming scenarios? I would expect that such trends due to a warming of the atmosphere are already visible in the datasets used here.

Are there limitations with respect to extreme events not present in the training data, e.g. with return times much longer than the available reanalysis and observations?

The authors describe in section 2.3, that a main advantage of the deep learning approach is that the entire spatial domain can be processed at once instead of using grid cell-wise univariate quantile mapping. This should also be an advantage when correcting spatial patterns. Therefore, I believe it would be insightful to test correlations or power spectral densities computed over single daily fields.

Since a main focus of the study here is on the use and benefits of attention layers in the architecture, the network should be compared against a baseline architecture that is fully convolutional without attention.

**Minor comments**

L57: ".. the upscaling aspect.." is upsampling meant here? Upscaling would mean a coarsening of the resolution.

L134: How large are the 25km and 5km resolution fields in terms of pixel-dimensions?

L134: "Zeroes in" might not be the best wording here.

L166: ".. and encoder and an encoder", the second one should be a decoder?

L176: Figure 2 instead of 3?

L207: Is the decoder or encoder trained here?

L235: Is empirical or parametric QM used?

L236: How many bins are used for QM? 15% bin width seems rather larger to me.

L461-463: As noted by the authors, stochasticity is crucial for modelling rainfall and uncertainty quantification with machine learning approaches. Since the network in this study includes dropout layers that can be used during inference for exactly this purpose, have you tried using them? It would be interesting to see the benefits.

L464: GANs are unsupervised and generally not part of reinforcement learning.

L469: To make this point, one should compare the network with attention used here against a fully convolutional network without attention layers.

Fig.1: Adding Fig.1c to Figure 3 would make the comparison much easier.

It would also greatly help the interpretation if all the result figures would include the model/dataset name as a subtitle.

Fig.3 Maybe add the color bar title as well?

Fig.7 Including a version of this figure without the log scale on the x-axis would improve the clarity of how the right tails are improved.

Fig.7 and 8. The color coding should be made consistent here to avoid confusion.

Tab.3. It would be important to see how correlation and RMSE are improved when computed between single days instead of mean precipitation only.